# Time and Frequency Domain Analysis of IMU-Based Orientation Estimation Algorithms with Comparison to Robotic Arm Orientation as Reference [note 1]

**DOI:** 10.3390/s25165161

**Published:** 2025-08-20

**Authors:** Ruslan Sultan, Steffen Greiser

**Affiliations:** Institute of Management and Technology, Hochschule Osnabrück, Kaiserstr. 10c, 49809 Lingen, Germany; s.greiser@hs-osnabrueck.de

**Keywords:** inertial measurement unit, orientation estimation, sensor fusion, robotic arm, time domain evaluation, frequency domain evaluation, composite frequency response, composite coherence

## Abstract

This work focuses on time and frequency domain analyses of IMU-based orientation estimation algorithms, including indirect Kalman (IKF), Madgwick (MF), and complementary (CF) filters. Euler angles and quaternions are used for orientation representation. A 6-DoF IMU is attached to a 6-joint UR5e robotic arm, with the robot’s orientation serving as the reference. Robotic arm data is obtained via an RTDE interface and IMU data via a CAN bus. Test signals include pose sequences, which are big-amplitude, slowly changing signals used to evaluate stationary and low-dynamics responses in the time domain, and small-amplitude, fast-changing generalized binary noise (GBN) signals used to evaluate dynamic responses in the frequency domain. To prevent poor filters’ performance, their parameters are tuned. In the time domain, RMSE and MaxAE are calculated for roll and pitch. In the frequency domain, composite frequency response and coherence are calculated using the Ockier method. RMSEs are computed for response magnitude and coherence, and averaged equivalent time delay (AETD) is derived from the response phase. In the time domain, MF and CF show the best overall performance. In the frequency domain, they again perform similarly well. IKF consistently performs the worst in both domains but achieves the lowest AETD.

## 1. Introduction

This paper is an extended version of our paper presented at the 10th International Conference on Sensors Engineering and Electronics Instrumentation Advances (SEIA’ 2024), Ibiza, Spain, 25–27 September 2024 [1].

Accurate orientation estimation is crucial for many applications and plays a huge role in aerospace, robotics, navigation, human movement analysis, and other areas [2,3,4]. Inertial measurement units (IMUs) are often used for orientation estimation. A 6-DoF (degree of freedom) IMU combines an accelerometer and a gyroscope that measure linear acceleration and angular velocity, respectively [2,4].

Orientation can be represented in several forms such as Euler angles, rotation matrices, axis-angle representation and quaternions [4,5,6] that describe the rotation required to align the sensor frame with the reference frame. In this work, Euler angles and quaternion orientation representations are used.

Euler angles describe three sequential rotations needed to arrive from the reference frame to the rotated target frame. In this work, ZY′X″ order of rotation is used. For this order of rotation, the first rotation happens around the Z axis of reference frame XYZ on angle *ψ* (yaw), the second rotation on angle *θ* (pitch) around the Y′ axis of frame X′Y′Z resulting from the first rotation, and the third rotation on angle *ϕ* (roll) around the X″ axis of frame X″Y′Z′ resulting from the second rotation. The resulting frame X″Y″Z″ corresponds to the target frame.

The body’s orientation can be estimated using an accelerometer. By measuring the projections of the gravity vector on the sensor axes, orientation can be estimated [2,3,4]. Gravity vector components in the body frame can be calculated using gravity vector components in the world reference frame pointing down, e.g., north-east-down frame (NED):(1)$$\begin{array}{c} g_{b}=R_{bw}\;g_{w}=R_{wb}^{-1}g_{w}=R_{wb}^{⊤}g_{w}=\\ ={\left(R_{z}\left(\psi \right)R_{y}\left(\theta \right)R_{x}\left(\phi \right)\right)}^{⊤}\left[\begin{array}{c} 0\\ 0\\ g \end{array}\right]=g\left[\begin{array}{c} -\sin\theta \\ \sin\phi \cos\theta \\ \cos\phi \cos\theta \end{array}\right] \end{array}$$
where $g_{b}$ is the gravity vector expressed in the body frame, $g_{w}={\left[\begin{array}{ccc} 0 & 0 & g \end{array}\right]}^{⊤}$ is the gravity vector expressed in the world frame (NED convention), $R_{bw}$ is the rotation matrix from world frame to body frame, $R_{wb}$ is the rotation matrix from body frame to world frame, $R_{wb}^{-1}=R_{wb}^{⊤}$ because rotation matrices are orthogonal, $R_{wb}=R_{z}\left(\psi \right)R_{y}\left(\theta \right)R_{x}\left(\phi \right)$ is constructed by successive ψ (yaw), θ (pitch), and ϕ (roll) rotations.

Using Equation (Equation 1), the roll and pitch angles can be estimated. However, the yaw angle cannot be determined by this method.(2)$$\phi =\arctan2(g_{y_{b}},g_{z_{b}})$$(3)$$\theta =\arcsin\left(-\frac{g_{x_{b}}}{g}\right)$$

Alternatively, body orientation can be estimated using a gyroscope. The relationship between body-frame angular velocities, which are measured by the gyroscope, and Euler angle rates and can be expressed as: (4)$$\left[\begin{array}{c} \omega_{xb}\\ \omega_{yb}\\ \omega_{zb} \end{array}\right]=T_{wb}\left(\phi ,\theta \right)\left[\begin{array}{c} \dot{\phi }\\ \dot{\theta }\\ \dot{\psi } \end{array}\right]\;\mathrm{or}\;\mathrm{equivalently}\;\left[\begin{array}{c} \dot{\phi }\\ \dot{\theta }\\ \dot{\psi } \end{array}\right]=T_{bw}^{-1}\left(\phi ,\theta \right)\left[\begin{array}{c} \omega_{xb}\\ \omega_{yb}\\ \omega_{zb} \end{array}\right],$$
where the transformation matrices are defined as: (5)$$T_{wb}\left(\phi ,\theta \right)=\left[\begin{array}{ccc} 1 & 0 & -\sin(\theta )\\ 0 & \cos(\phi ) & \cos(\theta )\sin(\phi )\\ 0 & -\sin(\phi ) & \cos(\theta )\cos(\phi ) \end{array}\right]$$(6)$$T_{bw}^{-1}\left(\phi ,\theta \right)=\left[\begin{array}{ccc} 1 & \sin(\phi )\tan(\theta ) & \cos(\phi )\tan(\theta )\\ 0 & \cos(\phi ) & -\sin(\phi )\\ 0 & \sin(\phi )/\cos(\theta ) & \cos(\phi )/\cos(\theta ) \end{array}\right]$$

By integrating the time derivatives of the Euler angles from Equation (Equation 4) over time and knowing the initial orientation, the current body orientation can be estimated [2,3]. For a discrete-time system with time step Δt, Euler angles at time *m* can be estimated using angles and rates at time m−1 using the forward Euler method:(7)$$\left[\begin{array}{c} \phi_{m}\\ \theta_{m}\\ \psi_{m} \end{array}\right]=\left[\begin{array}{c} \phi_{m-1}+{\dot{\phi }}_{m-1}\Delta t\\ \theta_{m-1}+{\dot{\theta }}_{m-1}\Delta t\\ \psi_{m-1}+{\dot{\psi }}_{m-1}\Delta t \end{array}\right]$$

The method of orientation estimation with a gyroscope is effective only for short periods of time [3]. Due to the noise, the estimated orientation drifts from the actual orientation over time, even if the body is stationary.

Orientation representation using quaternions avoids singularities such as gimbal lock, which are associated with Euler angles, while providing rotation-order independence (due to a single rotation representation instead of three sequential ones) and ensuring better numerical stability and computational efficiency. Rotation can be described by a unit quaternion [6]: (8)$$\begin{array}{c} q=q_{0}+q_{1}i+q_{2}j+q_{3}k,\\ \left(q_{0},\left(\begin{array}{c} q_{1}\\ q_{2}\\ q_{3} \end{array}\right)\right)=\left(\cos\frac{\alpha }{2},\;u\sin\frac{\alpha }{2}\right)\\ \alpha =2\cdot \arccos\left(q_{0}\right),\;i^{2}=j^{2}=k^{2}=-1,\\ \;q_{0},\ldots ,q_{3}\in R,\;\sqrt{q_{0}^{2}+q_{1}^{2}+q_{2}^{2}+q_{3}^{2}}=1, \end{array}$$
where usin(α/2) is a vector describing the rotation axis, and α is the rotation angle in the range [0,2π).

The calculation of orientation of quaternion components using gravity vector components is shown in [4]. The following system must be solved to determine the quaternion components [4]:(9)$$\left\{\begin{array}{cc} 2(q_{1}q_{3}+q_{0}q_{2}) & ={\hat{a}}_{xb}\\ 2(q_{2}q_{3}-q_{0}q_{1}) & ={\hat{a}}_{yb}\\ q_{0}^{2}-q_{1}^{2}-q_{2}^{2}+q_{3}^{2} & ={\hat{a}}_{zb}\\ \sqrt{{\hat{a}}_{xb}^{2}+{\hat{a}}_{yb}^{2}+{\hat{a}}_{zb}^{2}} & =1 \end{array}\right.$$
where ${\hat{a}}_{xb},{\hat{a}}_{yb},{\hat{a}}_{zb}$ are the normalized components of the measured acceleration vector in the body-fixed frame. Normalization is done by the magnitude of the measured acceleration vector. In static conditions, this magnitude equals the gravitational acceleration *g*. However, in dynamic conditions, the presence of linear accelerations causes the measured acceleration magnitude to deviate from *g*, which can introduce errors in the orientation estimation. The system (Equation 9) is underdetermined and can be solved only with additional constraints, such as assuming $q_{3}=0$ [4]. Under this assumption, the solution to the system is given by:(10)$$\left(\begin{array}{c} q_{0}\\ q_{1}\\ q_{2}\\ q_{3} \end{array}\right)=\left(\begin{array}{c} \sqrt{({\hat{a}}_{zb}+1)/2}\\ -{\hat{a}}_{yb}/\sqrt{2({\hat{a}}_{zb}+1)}\\ {\hat{a}}_{xb}/\sqrt{2({\hat{a}}_{zb}+1)}\\ 0 \end{array}\right)$$

Alternative solutions to system (Equation 9) are also presented in [4].

Calculation of orientation using angular rates in the body-fixed frame can also be written in quaternion form [2,4]:(11)$$\dot{q}=\frac{1}{2}\;q⊗\omega_{b}$$(12)$$\;\omega_{b}=\omega_{xb}\;i+\omega_{yb}\;j+\omega_{zb}\;k,$$
where ⊗ denotes the quaternion product between the orientation quaternion q and the quaternion representation of the angular velocity $\omega_{b}$ in the body-fixed frame.

Similar to Equation (Equation 11), the quaternion at time step *m* in a discrete-time system can be estimated using the forward Euler method: (13)$$q_{m}=q_{m-1}+{\dot{q}}_{m-1}\Delta t$$

Accelerometer and gyroscope data can be fused using different sensor fusion algorithms to achieve a more robust and accurate orientation estimation [3,4]. Various approaches have been proposed in the literature, each with specific advantages and drawbacks. Kalman filter variants, including the indirect Kalman filter (IKF) and the Extended Kalman Filter (EKF), offer a probabilistic framework and are theoretically optimal under Gaussian noise assumptions [2,3,4,7,8]. While the IKF reduces sensitivity to linearization errors compared to EKF by estimating only the error state, both approaches remain sensitive to sensor noise characteristics, rely on precise system modeling, and may experience degraded performance in highly dynamic conditions [2,3,4,7,8]. Complementary filters (CFs) [4] provide a computationally efficient alternative by blending high-frequency gyroscope data with low-frequency accelerometer information, yet they can lose accuracy in highly dynamic conditions as well. Gradient-descent-based methods, such as the Madgwick filter (MF) [2], offer fast convergence and good performance for low-cost IMUs without requiring a full dynamic model, but they may also accumulate orientation drift under fast rotational motions or significant transient linear accelerations affecting the accelerometer signal.

Using a 9-DoF IMU (accelerometer, gyroscope, magnetometer) allows for an orientation estimation relative to the Earth’s magnetic field and a gravity vector and helps correct gyroscope drift around the vertical axis [3,4]. However, a 9-DoF IMU is not used in this work because the experimental setup includes a robotic arm. Magnetometer measurements can be strongly affected by electromagnetic interference from its drives and other electrical equipment, as well as magnetic distortions caused by surrounding ferromagnetic materials.

To comprehensively evaluate the filters’ performance, traditional time-domain error metrics are combined with frequency-domain analysis, which captures key dynamic properties such as phase shift, magnitude response, and input–output coherence. While time-domain metrics like orientation error are effective for assessing accuracy during slow or static conditions, they may obscure important characteristics under dynamic motion, e.g., small time delays can result in large instantaneous errors despite accurate tracking. Frequency-domain analysis addresses these limitations by revealing how each filter attenuates or distorts signals across different frequency bands, quantifies phase delays, and measures linear correlation via coherence. These properties are robustly computed using the composite frequency response and coherence estimation approach described in the Ockier method [9].

Accurate computation of these metrics relies on a reliable reference for orientation. While indirect reference systems, such as optical tracking [10] or high-grade integrated inertial systems, can be used to provide such data, they may still lack the precision, repeatability, or long-term stability required to fully evaluate filter performance. In contrast, a high-precision robotic arm not only traces the pose with high accuracy but can also execute predefined trajectories reliably, enabling consistent and repeatable evaluation of orientation estimation algorithms. Several studies have used industrial robotic manipulators as precise, repeatable references for evaluating orientation or full pose estimation. Representative examples include Cavallo et al., who mounted an IMU on a KUKA manipulator to compare EKF-, Mahony-, and Madgwick-type estimators against the robot’s orientation [11]; El-Gohary and McNames, who validated joint-angle estimation with an IMU against a high-precision robot arm [12]; Ricci et al., who systematically characterized static and dynamic orientation accuracy using a KUKA LWR 4+ executing sinusoidal trajectories across frequencies [13]; Hislop et al. used an ABB IRB2600 industrial robot to validate the dynamic orientation accuracy of a commercial IMU system across several angular velocities and orientations, reporting error statistics under controlled motion conditions [14]; and more recently Kuti et al., who proposed a UR-series robot-based procedure that treats the manipulator as a high-accuracy reference for orientation sensors [15]. In line with this established practice, a robot arm is used as reference in this work. However, in contrast to most prior work that primarily reports time-domain metrics, our study adds an explicit frequency-domain evaluation, an analysis that remains relatively uncommon in the literature.

The remainder of this paper is structured as follows. Section 2 introduces the orientation estimation filters considered in this study. Section 3 describes the experimental setup, including the IMU and robotic arm configuration. Section 4 and Section 5 present the evaluation methodology and results in the time and frequency domains, respectively. Finally, Section 6 discusses the findings and concludes the paper.

## 2. Filters

### 2.1. Indirect Kalman Filter

The Indirect Kalman Filter (IKF), also known as the Error State Kalman Filter (ESKF), is a variant of the traditional Kalman Filter. In IKF, the state vector represents the state error rather than the actual state itself [7,8]. IKF is mathematically equivalent to a standard KF for linear systems, but it offers practical advantages in nonlinear settings [7,8]. In particular, by tracking the linearization error explicitly, the IKF often yields better numerical stability and accuracy than a conventional extended Kalman Filter (EKF) when estimating orientation [7,8].

In this work, IKF from the Sensor Fusion and Tracking Toolbox of MATLAB is used [16]. It uses a 9-element error state that tracks the orientation error, gyro bias error, and linear acceleration error [16]. Kalman equations used in this algorithm [16]: (14)$$\begin{array}{cc} x_{k}^{-} & =0 \end{array}$$(15)$$\begin{array}{cc} P_{k}^{-} & =Q_{k} \end{array}$$(16)$$\begin{array}{cc} y_{k} & =z_{k} \end{array}$$(17)$$\begin{array}{cc} S_{k} & =R_{k}+H_{k}P_{k}^{-}H_{k}^{⊤} \end{array}$$(18)$$\begin{array}{cc} K_{k} & =P_{k}^{-}H_{k}^{⊤}S_{k}^{-1} \end{array}$$(19)$$\begin{array}{cc} x_{k}^{+} & =K_{k}y_{k} \end{array}$$(20)$$\begin{array}{cc} P_{k}^{+} & =P_{k}^{-}-K_{k}H_{k}P_{k}^{-} \end{array}$$
where $x_{k}^{-}$ is the predicted state estimate representing the error process, $P_{k}^{-}$ is the predicted estimate covariance, $y_{k}$ is the innovation (or residual), $S_{k}$ is the innovation covariance, $K_{k}$ is the Kalman gain, $x_{k}^{+}$ is the updated (corrected) state estimate, and $P_{k}^{+}$ is the updated estimate covariance.

MATLAB’s IKF has the following parameters: variance in accelerometer signal noise, variance in gyroscope signal noise, variance in gyroscope offset drift, variance in linear acceleration noise, and decay factor for linear acceleration drift [16]. In MATLAB’s implementation of the IKF, the decay factor for linear acceleration drift is a scalar parameter in the range [0, 1), used to model the drift in linear acceleration as a low-pass-filtered white noise process [16]. It controls how quickly the estimated linear acceleration bias decays over time. A lower value causes the filter to adapt more rapidly to changes in acceleration, which is suitable for highly dynamic motion. Conversely, a higher value assumes more slowly varying acceleration and results in a more persistent bias estimate. To improve filter performance, default parameters of IKF were tuned (All filters used in this work were tuned using a Generalized Binary Noise (GBN) signal with a wider frequency range than the test signal GBN to introduce a broader range of dynamic conditions. The description of GBN is provided in Section 5.1.3) by using MATLAB’s tune function with a custom cost function being the sum of root mean squared errors (RMSEs) for roll and pitch. In general, RMSE can be calculated as: (21)$$\mathrm{RMSE}=\sqrt{\frac{1}{n}\sum_{i=1}^{n}{\left(y_{i}-{\hat{y}}_{i}\right)}^{2}}$$
where $y_{i}$ is the true value, ${\hat{y}}_{i}$ is the predicted value, and *n* is the number of observations.

### 2.2. Complementary Filter

In this work, a quaternion-based complementary filter from Sensor Fusion and Tracking Toolbox of MATLAB [17] is used. Its implementation is described in [4]. In this filter, the gyroscope’s integrated orientation provides the high-frequency component of motion (capturing rapid rotations), while the accelerometer provides a low-frequency reference from the gravity vector (stabilizing long-term drift) [4,18].

The complementary filter blends the orientation estimates from the gyroscope and accelerometer using a weighted average, followed by normalization to ensure the result remains a unit quaternion. The update equation is given by [4]: (22)$$q_{k}=\frac{\alpha \;q_{\mathrm{gyro},k}+\left(1-\alpha \right)\;q_{\mathrm{acc},k}}{∥\alpha \;q_{\mathrm{gyro},k}+\left(1-\alpha \right)\;q_{\mathrm{acc},k}∥},$$
where $q_{\mathrm{gyro},k}$ is the orientation estimate obtained by integrating the gyroscope measurements, $q_{\mathrm{acc},k}$ is the estimate derived from the accelerometer, and α∈ [0, 1] is the filter gain weighting the gyroscope contribution.

The choice of gain α significantly affects the filter dynamics. A larger value of (1−α), i.e., a higher accelerometer gain, causes the filter to converge more rapidly to the gravity-based orientation but increases sensitivity to noise and transient disturbances. Conversely, a smaller value of (1−α) results in greater responsiveness to rapid motion but slower drift correction.

To improve filter performance, the default accelerometer gain (1−α)=0.01 was tuned by grid search to minimize the sum of RMSEs for roll and pitch on the tuning signal. The tuned gain is 0.005.

### 2.3. Madgwick Filter

The Madgwick filter is a quaternion-based attitude estimator that fuses gyroscope and accelerometer data to track orientation. Originally introduced by Madgwick [2], it uses a single scalar gain β to balance gyroscope integration against accelerometer-based correction. The algorithm maintains a unit quaternion q representing orientation. At each timestep, the filter integrates the angular velocity ω to predict the new orientation, then corrects it by a gradient descent step that aligns the estimated gravity direction with the measured accelerometer vector [2].

The quaternion update can be summarized as [2]:(23)$$\begin{array}{cc} {\dot{q}}_{\omega } & =\frac{1}{2}\;q⊗{\left[0,\omega \right]}^{T}, \end{array}$$(24)$$\begin{array}{cc} {\dot{q}}_{\mathrm{est}} & ={\dot{q}}_{\omega }-\beta \frac{\nabla f(q)}{∥\nabla f(q)∥}, \end{array}$$(25)$$\begin{array}{cc} q & \leftarrow q+{\dot{q}}_{\mathrm{est}}\;\Delta t, \end{array}$$
followed by normalization to unit length.

The filter represents orientation by a unit quaternion q. Gyroscope measurements $\omega ={\left[\omega_{x},\omega_{y},\omega_{z}\right]}^{T}$ are converted to a quaternion derivative ${\dot{q}}_{\omega }=\frac{1}{2}q⊗{\left[0,\omega \right]}^{T}$. Pure gyroscope integration causes drift. Hence, the accelerometer $a={\left[a_{x},a_{y},a_{z}\right]}^{T}$ is used to correct this drift.

After normalization, $\hat{a}=a/∥a∥$, the filter minimizes the error: (26)$$f\left(q\right)=q^{*}⊗\hat{g}⊗q-{\left[0,\hat{a}\right]}^{T},$$
where $\hat{g}={\left[0,0,0,1\right]}^{T}$ is the global gravity vector. The filter derives the gradient analytically:(27)$$\nabla f\left(q\right)=J_{g}{\left(q\right)}^{T}f_{g}\left(q\right),$$
where: (28)$$\begin{array}{cc} f_{g}\left(q\right) & =\left[\begin{array}{c} 2\left(q_{1}q_{3}-q_{0}q_{2}\right)-a_{x}\\ 2\left(q_{0}q_{1}+q_{2}q_{3}\right)-a_{y}\\ 2\left(q_{0}^{2}-q_{1}^{2}-q_{2}^{2}+q_{3}^{2}\right)-a_{z} \end{array}\right], \end{array}$$(29)$$\begin{array}{cc} J_{g}\left(q\right) & =\left[\begin{array}{cccc} -2q_{2} & 2q_{3} & -2q_{0} & 2q_{1}\\ 2q_{1} & 2q_{0} & 2q_{3} & 2q_{2}\\ 0 & -4q_{1} & -4q_{2} & 0 \end{array}\right]. \end{array}$$

Madgwick’s algorithm requires only a single gradient descent step per update [2]. The gradient ∇f/∥∇f∥ defines the correction axis. The correction is subtracted from the gyro-integrated quaternion derivative, thereby reducing tilt errors caused by gyroscope drift. To summarize, the main steps of the MF are:Normalize accelerometer: $\hat{a}=a/∥a∥$.Compute gyro quaternion derivative: ${\dot{q}}_{\omega }=\frac{1}{2}q⊗{\left[0,\omega \right]}^{T}$.Compute ∇f(q) using Jacobian $J_{g}$ and error $f_{g}$.Normalize gradient: g=∇f/∥∇f∥.Apply correction: ${\dot{q}}_{\mathrm{est}}={\dot{q}}_{\omega }-\beta g$.Integrate quaternion: $q_{\mathrm{new}}=q+{\dot{q}}_{\mathrm{est}}\;\Delta t$.Normalize $q_{\mathrm{new}}$.

The scalar β is the only tuning parameter in the Madgwick filter. It controls how strongly the accelerometer correction is applied. A larger β results in faster convergence but increased sensitivity to noise. The relationship between β and the expected gyroscope measurement noise $\omega_{\mathrm{error}}$ is:(30)$$\beta =\sqrt{\frac{3}{4}}\omega_{\mathrm{error}}.$$

In practice, β can be tuned to improve filter performance. Grid search for β was used to minimize the sum of RMSEs for roll and pitch on the tuning signal. The resulting β used in this work is 0.006.

## 3. Experimental Setup

To evaluate the estimated orientation, it is essential to establish a reference orientation. Modern robotic arms, equipped with six or more joints, enable complex, precise, and accurate dynamic motion of an end effector, such as a gripper. For this reason, in this work, a Bosch MM7.10 IMU (Bosch Rexroth AG, Lohr am Main, Germany) is mounted on a 6-joint UR5e robotic arm (Universal Robots A/S, Odense, Denmark) to accurately track IMU orientation accurately (pose repeatability per ISO 9283 [19]: ±0.03 mm [20]) and execute dynamic maneuvers (maximum velocity: 4 m/s [20]). The experimental setup is shown in Figure 1.

The UR5e controller provides robot data such as the pose of the tool center point (TCP) frame via the TCP/IP-based (Here, TCP/IP refers to the Transmission Control Protocol/Internet Protocol, not to the tool center point) Real-Time Data Exchange (RTDE) interface with a configurable rate up to 500 Hz [21]. In this work, a rate of 100 Hz is used. For working with RTDE interface, the RTDE Python Client Library v2.7.2 is used.The TCP pose vector contains the coordinates and orientation of the TCP frame in the base frame (used as a reference frame) of the UR5e robot. Orientation of the TCP frame is provided in the rotation vector form (term used by Universal Robots for axis–angle representation), which is then transformed to quaternion and Euler angles representations.

The Bosch MM7.10 IMU provides accelerometer and gyroscope measurements at a rate of 100 Hz via the Controller Area Network (CAN) bus. The performance characteristics for angular rate and acceleration measurements, evaluated at room temperature, are summarized in Table 1 [22]. Our experiments were also carried out under similar room temperature conditions. The IMU frame is aligned with the TCP frame of the robot with a small offset along the Z-axis (around four centimetres). At the start of an experiment, TCP, and thus IMU, frames are aligned with the base frame. A PCAN-USB adapter (PEAK-System Technik GmbH, Darmstadt, Germany) provides the physical connection between the IMU and the computer, while data acquisition is handled using the Python-CAN v4.2.2 library.

After recording the robot and IMU data (both timestamped using the recording computer’s system clock), the two datasets were merged by linearly interpolating values to a common set of timestamps. Due to transmission and buffering delays there is a mismatch between the IMU and robot timestamps. To compensate for this, the time shift between the IMU and robot data was estimated by minimizing the Averaged Equivalent Time Delay (AETD) (for more details on the AETD calculation, see Equation (Equation 41)), based on the composite frequency response phase, with the time-shifted robot orientation as input and the integrated gyroscope measurements as output. The time shift that resulted in the lowest AETD was selected. For the pose sequence test signals (see description in Section 4.1.1), the following shifts were estimated and compensated: signal 1—31 ms; signal 2—16 ms; signal 3—20 ms. For the Generalized Binary Noise test signals (see description in Section 5.1.3): GBN roll—26 ms; GBN pitch—82 ms; GBN roll and pitch—4 ms.

For data postprocessing, filter implementation, parameter tuning and performance evaluation of the filters, MATLAB 2024a is used. For the evaluation of filter performance in the frequency domain, the MATLAB tool FitlabGui [9] developed by the DLR (German Aerospace Center) is used.

## 4. Time Domain Evaluation

### 4.1. Description and Methodology

#### 4.1.1. Pose Sequence Signal

For the time-domain evaluation, pose sequence test signals are used, where pose sequence refers to a big-amplitude slowly changing (compared to small-amplitude, fast-changing signals such as GBN (Section 5.1.3)) sequential change in the robot’s pose over time. These signals are used in this work to assess the filters’ stationary behaviour and low-dynamic responses in the time domain. In pose sequence test signals 1 and 2, both the orientation and position of the TCP are varied to evaluate the filters’ performance under complex motion (translation and rotation). Pose sequence test signal 3 involves only a change in orientation (rotation around the TCP), without any position change. It is important to note that even pure rotation around the TCP results in the norm of the acceleration measured by the accelerometer not being equal to *g*, because the rotation does not occur around the accelerometer’s own frame (i.e., the IMU frame). The lever arm between the TCP and IMU frames introduces tangential and centripetal accelerations that are sensed by the accelerometer. The influence of this rotational motion on the linear acceleration at the IMU frame is given by: (31)$$a_{\mathrm{IMU}}=a_{\mathrm{TCP}}+\underset{\mathrm{tangential}\;\mathrm{acceleration}}{\underset{︸}{\alpha \times r_{\mathrm{IMU}/\mathrm{TCP}}}}+\underset{\mathrm{centripetal}\;\mathrm{acceleration}}{\underset{︸}{\omega \times \left(\omega \times r_{\mathrm{IMU}/\mathrm{TCP}}\right)}}$$
where $a_{\mathrm{IMU}}$ and $a_{\mathrm{TCP}}$ are the linear accelerations of the IMU and the TCP, respectively, α is the angular acceleration, ω is the angular velocity, and $r_{\mathrm{IMU}/\mathrm{TCP}}$ is the position vector from the TCP to the IMU.

#### 4.1.2. Metrics

Root mean squared error (RMSE) is calculated for roll and pitch angles. Additionally, the maximum absolute error (MaxAE) is used.

### 4.2. Results

The RMSE and MaxAE values for roll and pitch angles and the sum of the errors for the angles, corresponding to the used filter and test signal, are presented in Table 2. The lowest value for a given test signal and metric is marked in green, and the biggest is in red. Errors for yaw angle are not taken into account due to the uncompensated yaw drift. Yaw drift happens due to the gyroscope drift and the inability of the gravity components analysis method to estimate the yaw angle and thus compensate for the drift. For the same reason, the cost function for tuning filters only included RMSEs for roll and pitch.

Figure 2 shows the robot (reference) orientation vs. estimated orientation for pose sequence test signal 1. Figure 3 shows the robot (reference) orientation vs. estimated orientation for pose sequence test signal 2. Figure 4 shows the robot (reference) orientation vs. estimated orientation for pose sequence test signal 3.

Considering the sum of RMSEs for roll and pitch, MF achieves the best performance in two out of three test signals. When evaluating the sum of MaxAEs for roll and pitch, CF performs best in two of the three test signals. Overall, the performance of CF and MF is very similar (the largest differences are 0.38° in the sum of RMSEs and 0.87° in the sum of MaxAEs), while IKF shows the worst performance (with the largest differences being 2.2° in the sum of RMSEs and 5.44° in the sum of MaxAEs).

## 5. Frequency Domain Evaluation

### 5.1. Description and Methodology

#### 5.1.1. Discrete Fourier Transform

An important part of the evaluation in the frequency domain is the Fourier Transform, which provides the frequency spectrum of the signal.

By means of the discrete-time Fourier transform (DTFT), the frequency content of infinite-length discrete-time signals can be analyzed. It produces a continuous function of frequency defined over the interval [−π,π] and is used primarily for the analysis of linear time-invariant (LTI) systems and signal behavior [23]. DTFT of a discrete-time signal x[n] is defined as:(32)$$X\left(e^{j\omega }\right)=\sum_{n=-\infty }^{\infty }x\left[n\right]e^{-j\omega n},\;\omega \in \left[-\pi ,\pi \right]$$

The discrete Fourier transform (DFT) is a sampled version of the DTFT, applied to finite-length discrete-time signals [23]. Given a sequence x[n], where n=0,1,…,N−1, the DFT transforms it into a sequence X[k] of the same length, representing frequency components. The DFT is defined as [23]:(33)$$X\left[k\right]=\sum_{n=0}^{N-1}x\left[n\right]\cdot e^{-j\frac{2\pi }{N}kn},\;\mathrm{for}\;k=0,1,\ldots ,N-1$$
where X[k] represents the complex amplitude of the frequency component at index *k*, and $j=\sqrt{-1}$ is the imaginary unit.

DFT is commonly implemented via the Fast Fourier Transform (FFT), an efficient algorithm for computing the frequency content of discrete-time signals. However, the FFT achieves peak performance only when the input size is a power of two or composed of small prime factors. For data lengths involving large primes, the FFT becomes inefficient due to its dependency on favorable factorization of the input length [23]. In contrast, the Chirp Z-Transform (CZT) offers more flexibility and efficiency in such cases, since its algorithmic structure does not rely on the factorization of the signal length [24].

Moreover, while the FFT computes the frequency response at uniformly spaced points on the unit circle in the complex plane, the CZT allows evaluation along arbitrary contours. This makes it ideal for focused spectral analysis where only a specific frequency band is of interest, and where increased frequency resolution is required without increasing the time-domain sample size [25]. The CZT can thus perform a zoomed-in frequency analysis more efficiently than zero-padding and interpolating an FFT.

CZT of a discrete-time signal $x_{n}$ is defined as the evaluation of its Z-transform at a sequence of complex points $z_{k}$. The general form of the CZT is given by [24]:$$X_{k}=\sum_{n=0}^{N-1}x_{n}z_{k}^{-n},\;k=0,1,\ldots ,M-1$$
where $x_{n}$ is a signal of length *N*, and $z_{k}\in C$ are the points in the complex plane (Z-plane) at which the transform is evaluated. These points are typically chosen to lie along a spiral contour defined by $z_{k}=AW^{-k}$, where *A* is the starting point and *W* is the ratio between successive points.

The Discrete Fourier Transform (DFT) is a special case of the CZT, obtained when the $z_{k}$ are uniformly spaced around the unit circle [24]:$$z_{k}=\exp\left(j\frac{2\pi }{N}k\right),\;k=0,1,\ldots ,N-1$$

#### 5.1.2. Windowing

DFT assumes that the input finite-duration signal represents one period of a periodic signal [23]. This means that the signal is conceptually extended periodically, repeating every *N* samples. If the values at the start and end of the segment do not match in amplitude and slope, this creates a discontinuity in the periodic extension. Such discontinuities introduce high-frequency artifacts that spread energy across multiple frequency bins, a phenomenon known as spectral leakage [26]. Spectral leakage obscures the true frequency content of the signal and degrades the spectral estimate.

To reduce spectral leakage, window functions (e.g., Hanning or Hamming windows) are commonly applied before computing the DFT. These functions taper the signal smoothly to zero at the segment boundaries, thereby minimizing the discontinuity and improving the frequency representation [23]. Figure 5 demonstrates how a Hanning window mitigates spectral leakage introduced by discontinuity. A 10 Hz sine wave with a sampling rate of 100 Hz was used in this example.

#### 5.1.3. Generalized Binary Noise

A generalized binary noise (GBN) signal was used in this work for filters’ performance evaluation and additionally for tuning filters’ parameters.

A step signal contains predominantly low frequencies [27]. A pseudo-random binary sequence (PRBS) contains the entire frequency range with signal energy equally distributed across all frequencies [27]. PRBS randomly switches with 0.5 probability between low and high levels of the same amplitude [27,28]; in other words, it comprises step signals with randomly chosen durations. Using GBN introduced by Tulleken [28] with the method presented in [27] allows for the excitation of frequencies in the given range. Similar to PRBS, GBN also randomly switches between low and high levels with the same amplitude [27,28]. Unlike PRBS, which switches randomly between levels with 0.5 probability, the probability of switching for GBN can vary [27,28]. The calculation of the non-switching probability $P_{ns}$ for the given frequency range and switching time is presented in Equation (Equation 34) from [27]. If $P_{ns}$ is equal to 1, then GBN behaves like a step signal, and if $P_{ns}$ is equal to 0.5, then GBN behaves like PRBS [27,28]. The upper-frequency limit for GBN can be defined by the Nyquist frequency, i.e., $\pi /T_{sw}$ [27,28]. To increase the upper-frequency limit, it is important to reduce switching time. In this work, it is achieved by using small amplitudes and moving the robot with high velocity and acceleration. For the GBN implementation on UR5e robot, the following settings were chosen: tool speed: 3m/s, tool acceleration: $15\;m/s^{2}$.(34)$$P_{ns}=\frac{1}{1+\sqrt{\tan\left(\frac{\omega_{L}T_{sw}}{2}\right)\tan\left(\frac{\omega_{H}T_{sw}}{2}\right)}}$$
where $P_{ns}$ is the non-switching probability, $\omega_{L}$ and $\omega_{H}$ are the lower and upper bounds of the frequency interval, and $T_{sw}$ is the switching time.

It is also worth noting that the frequency spectrum of GBN is not strictly confined to the interval $\omega_{L}$ to $\omega_{H}$; rather, most of the signal power is concentrated in that region [27,28].

GBN signal was used for tuning filters’ parameters. It was applied to roll and pitch axes with an amplitude of 3° for the frequency range of 0.1–25 rad/s, duration of 60 s, and empirically estimated $T_{sw}$ equal to 0.12 s (corresponding $P_{ns}$ value of 0.77).

GBN was also used as a test signal. It was applied to roll and pitch axes (separately and together) with an amplitude of 5° for the frequency range of 0.5–20 rad/s, duration of 30 s, and empirically estimated $T_{sw}$ equal to 0.15 s (corresponding $P_{ns}$ value of 0.58). This GBN test signal is shown in Figure 6. Compared to the GBN signal for tuning, the GBN signal for testing has a smaller frequency range, shorter duration, bigger amplitude, and bigger $T_{sw}$.

#### 5.1.4. Composite Frequency Response

For filters’ evaluation in the frequency domain, composite frequency response and coherence are used in this work. The composite frequency response and coherence are calculated using the Ockier method described in [9].

The frequency response of an LTI system describes how the system modifies the amplitude and phase of sinusoidal components of an input signal. Mathematically, it is defined as the Fourier transform of the system’s impulse response h(t) [29]:$$H\left(f\right)=\int_{-\infty }^{\infty }h\left(t\right)e^{-j2\pi ft}dt$$

In the discrete-time case, the frequency response is the Discrete-Time Fourier Transform (DTFT) of the impulse response [29]:$$H\left(e^{j\omega }\right)=\sum_{n=-\infty }^{\infty }h\left[n\right]e^{-j\omega n}$$

The magnitude |H(f)| describes how much each frequency is amplified or attenuated, and the angle ∠H(f) describes the phase shift introduced at each frequency.

In practical cases, a system is often unknown and estimated based on input–ouput signals (so-called system identification). The frequency response can be estimated from input–output data using power spectral densities [9,30]:(35)$$\hat{H}\left(f\right)=\frac{S_{xy}\left(f\right)}{S_{xx}\left(f\right)}$$
where $S_{xy}\left(f\right)$ is the cross-spectral density (cross-spectrum) between x(t) and y(t), and $S_{xx}\left(f\right)$ is the auto-spectral density (auto-spectrum) of x(t).

The auto-spectral densities of the input and output signals are given by [9]:(36)$$S_{xx}\left(f_{k}\right)=\frac{2}{T}{\left|X(f_{k})\right|}^{2}$$(37)$$S_{yy}\left(f_{k}\right)=\frac{2}{T}{\left|Y(f_{k})\right|}^{2}$$

The cross-spectral density between input and output is defined as [9]:(38)$$S_{xy}\left(f_{k}\right)=\frac{2}{T}X^{*}\left(f_{k}\right)Y\left(f_{k}\right)$$
where $X(f_{k})$, $Y(f_{k})$ are the Fourier transforms of x(t) and y(t) evaluated at frequency $f_{k}$, *T* is the duration of the time window, and * denotes the complex conjugate.

The magnitude-squared coherence function $\gamma_{xy}^{2}\left(f\right)$ measures the linear correlation between input and output at each frequency and is defined as [9,31]:(39)$$\gamma_{xy}^{2}\left(f\right)=\frac{|S_{xy}{\left(f\right)|}^{2}}{S_{xx}\left(f\right)S_{yy}\left(f\right)}$$

The coherence ranges from zero (no correlation) to one (perfect linear relationship at that frequency) [31].

The composite frequency response extends the classical concept by combining multiple estimates obtained from windowed data segments, thereby reducing random variability and improving reliability in practical system identification [9]. The data is first segmented, and a window function (e.g., Hanning or Hamming) is applied to each segment before estimating its frequency response. Averaging these individual windowed-segment (hereafter referred to as ’window’) responses reduces random error and increases the resolution of the amplitude and phase of the frequency response [9,31].

However, segmentation increases the fundamental frequency, i.e., the lowest frequency detectable by DFT, and decreases the frequency resolution of DFT. In conclusion, bigger windows are better for the evaluation of lower frequencies, and smaller windows are better for the evaluation of high frequencies. Thus, using different window sizes and consequently number of windows makes it possible to obtain a more accurate frequency response and coherence. Ockier shows the empirically determined frequency ranges (normal and high resolution) that can be reliably identified for a given number of windows and fundamental frequency. For normal resolution (used in this work) [9]:(40)$$f_{1}\left[{\left(\frac{n_{d}}{10}\right)}^{2}+1\right]<f_{k}<f_{1}\left[{\left(\frac{n_{d}}{10}\right)}^{2}+1+14\frac{n_{d}}{10}\right]$$
where $f_{1}$ is the fundamental frequency, and $n_{d}$ is the number of windows into which the data record is divided.

The process of calculating a composite frequency response and coherence with the Ockier method for each number of segments (windows) involves the following steps [9,31,32]:Segmenting the input and output signals into the segments that have 50% overlap,Windowing each segment with Hanning windows to reduce spectral leakage,Applying DFT, more specifically Chirp-Z transform, on each window,Calculation of auto- and cross spectral densities across the windows,Calculation of the frequency response and coherence based on the auto- and cross-spectral densities using Multiple Input Single Output (MISO) conditioning,

After calculating the frequency responses and coherences for each number of windows using MISO conditioning, they are weighted at each frequency point using weights based on Equation (Equation 40), and the composite frequency response and coherence are then calculated [9].

#### 5.1.5. Metrics

As a metric for frequency domain evaluation, the RMSE of the composite frequency response magnitude (RMSEM) is calculated with 0 dB being the reference magnitude. Moreover, the RMSE of the composite coherence (RMSEC) is calculated where a coherence of one represents the desired value, indicating a linear behaviour between the UR5e orientation and the estimated orientation. Additionally, based on the composite frequency response phase, the averaged equivalent time delay (AETD) is calculated according to:(41)$$AETD=-\frac{1}{n}\sum_{i=0}^{n}\frac{\gamma_{i}}{\omega_{i}}$$
where $\gamma_{i}$ is the phase of frequency response corresponding to the angular frequency $\omega_{i}$, *n* is the number of frequencies in the frequency response.

#### 5.1.6. Evaluation Example

As an example for evaluation, a second-order transfer function with a known frequency response is considered. In the Laplace domain, the transfer function of the system can be written as [33]:(42)$$H\left(s\right)=\frac{\omega_{n}^{2}}{s^{2}+2\zeta \omega_{n}s+\omega_{n}^{2}}$$
where $\omega_{n}$ is the natural frequency of the system in radians per second, and ζ is the damping ratio.

For this example, $\omega_{n}=10\;\mathrm{rad}/s$ and ζ=0.4 are selected. The system is evaluated over the frequency range from 0.1 to 100 rad/s. A sine chirp signal with logarithmic frequency progression in the same range (0.1 to 100 rad/s) is applied as input (Figure 7a), and the known frequency response (model-based) is compared with the composite frequency response identified from the input and output signals using the Ockier method. As shown in Figure 7b, the two responses are nearly identical.

Subsequently, the evaluation frequency range (0.1 to 100 rad/s) is retained, while the input chirp signal frequency range is reduced to 0.1 to 50 rad/s (Figure 8a). As expected, the composite frequency response (Figure 8b) is accurately identified only within this narrower excitation range.

Figure 9 shows a comparison of the composite coherence obtained from the chirp input covering the full frequency range of interest versus the chirp input applied over a shorter frequency range. As expected, the composite coherence of the full-range chirp remains close to one across the entire frequency range of interest, whereas the coherence of the shorter-range chirp drops close to zero beyond its excitation range.

### 5.2. Results

The frequency domain evaluation was done in the range of 0.5–30 rad/s. The lower frequency limit was chosen based on the longest window duration, while the upper limit was set slightly above the highest frequency (20 rad/s) of the GBN test signal because the frequency spectrum of GBN is not strictly within interval $\omega_{L}$ to $\omega_{H}$ from Equation (Equation 34).

The GBN test signal was applied to the roll and pitch axes separately and simultaneously. The GBN test signal applied separately to the roll and pitch axes is shown in the time domain in Figure 10a,b.

Composite frequency response (magnitude and phase) and composite coherence for GBN test signal applied separately to the roll and pitch axes is shown in Figure 11. Figure 12 shows the composite frequency response and composite coherence for the GBN test signal applied simultaneously to the roll and pitch axes.

Table 3 shows the RMSE of the composite frequency response magnitude, the RMSE of the composite coherence and the averaged equivalent time delay. In the table, the lowest value for a given test signal and metric is marked in green, and the biggest is in red. For the GBN test signal applied simultaneously to the roll and pitch axes, the reported metrics correspond to the sum of the individual metrics for roll and pitch.

Based on the RMSEM (Table 3), MF achieves the best results across all three test signals. CF delivers a performance similar to MF, with the largest difference in RMSEM being only 0.04 dB. RMSEC is close to zero for all three filters, and the differences between them are negligible (the largest difference is only 0.002). IKF shows the worst performance in terms of RMSEM. However, it achieves the best AETD across all three test signals (with the largest difference being 10 ms).

## 6. Conclusions and Discussion

In this work, three IMU orientation estimation algorithms: IKF, CF, and MF, were evaluated in the time domain using pose sequence test signals and in the frequency domain using GBN signals. For big-amplitude, slowly changing pose sequence signals (stationary and low-dynamics responses as observed in the time domain), MF and CF demonstrated the best performance. Similarly, for small amplitude, fast-changing signals such as GBN (dynamic responses as observed in the frequency domain), MF and CF again demonstrated the best performance with very similar results. IKF overall performed the worst in both the time and frequency domains, possibly due to overfitting during tuning. Despite this, IKF achieved the lowest AETD.

In this work, the orientation of a robotic arm was utilized as the reference orientation. The evaluation was conducted in both the time and frequency domains, with specific test signals and metrics defined for each. This testing approach has the potential to be extended beyond orientation estimation to include position estimation or both (pose) and may also be valuable for evaluating other sensors and fusion algorithms.

## Figures and Tables

**Figure 1 sensors-25-05161-f001:**
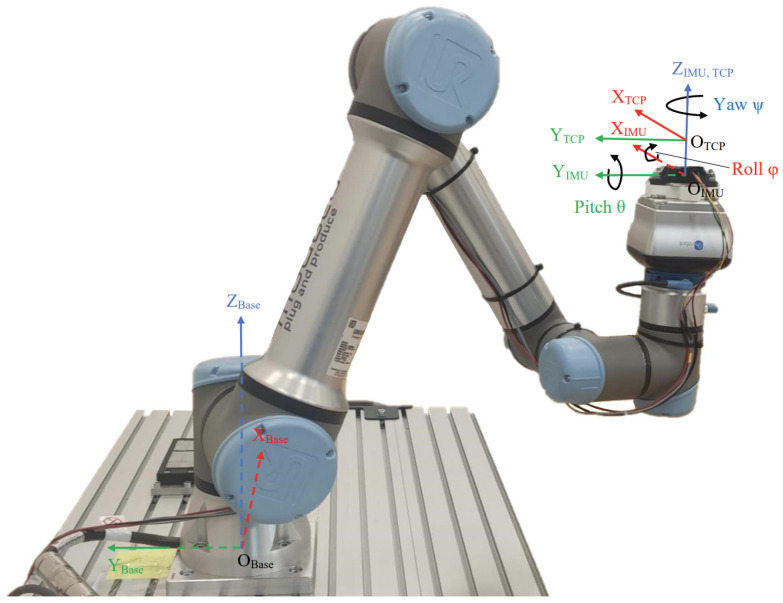
Experimental setup.

**Figure 2 sensors-25-05161-f002:**
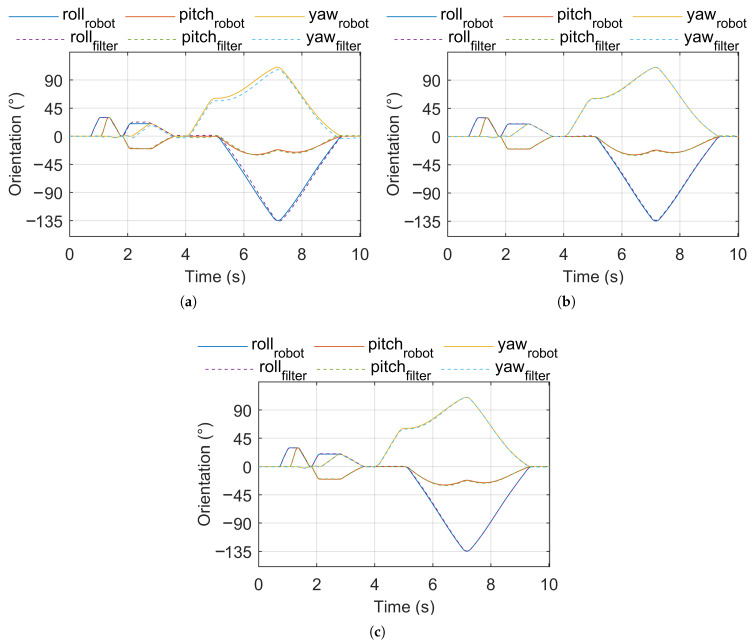
Reference vs. estimated orientation, pose sequence test signal 1. (**a**) IKF. (**b**) CF. (**c**) MF.

**Figure 3 sensors-25-05161-f003:**
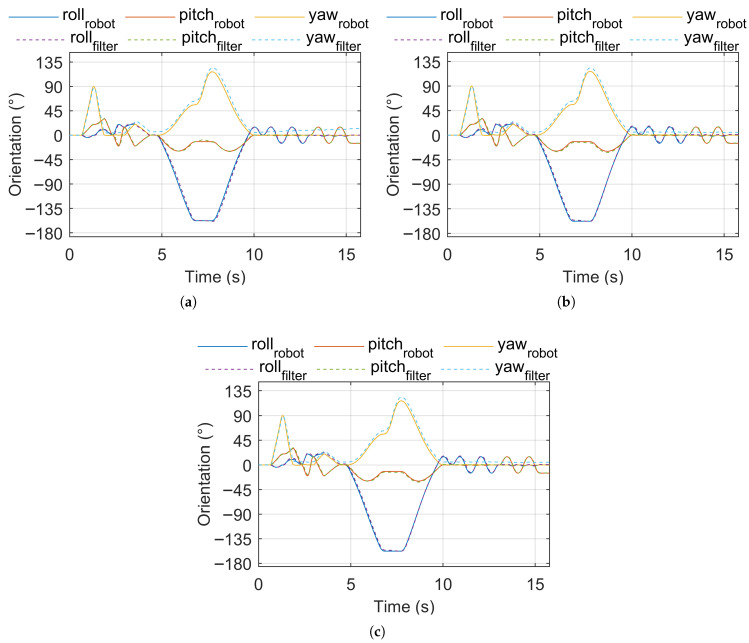
Reference vs. estimated orientation, pose sequence test signal 2. (**a**) IKF. (**b**) CF. (**c**) MF.

**Figure 4 sensors-25-05161-f004:**
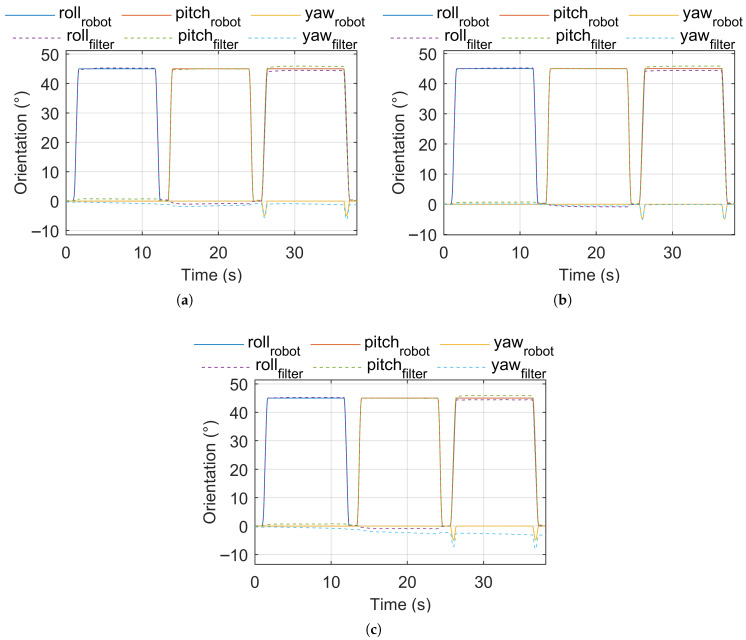
Reference vs. estimated orientation, pose sequence test signal 3. (**a**) IKF. (**b**) CF. (**c**) MF.

**Figure 5 sensors-25-05161-f005:**
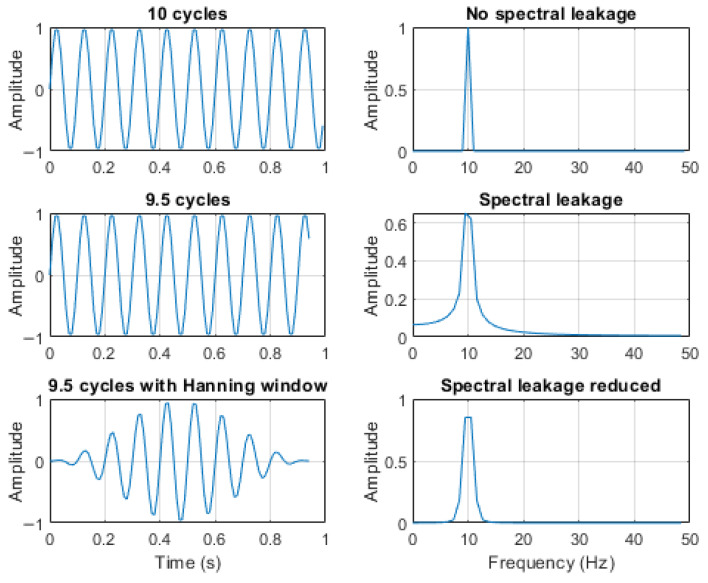
Effect of windowing on spectral leakage.

**Figure 6 sensors-25-05161-f006:**
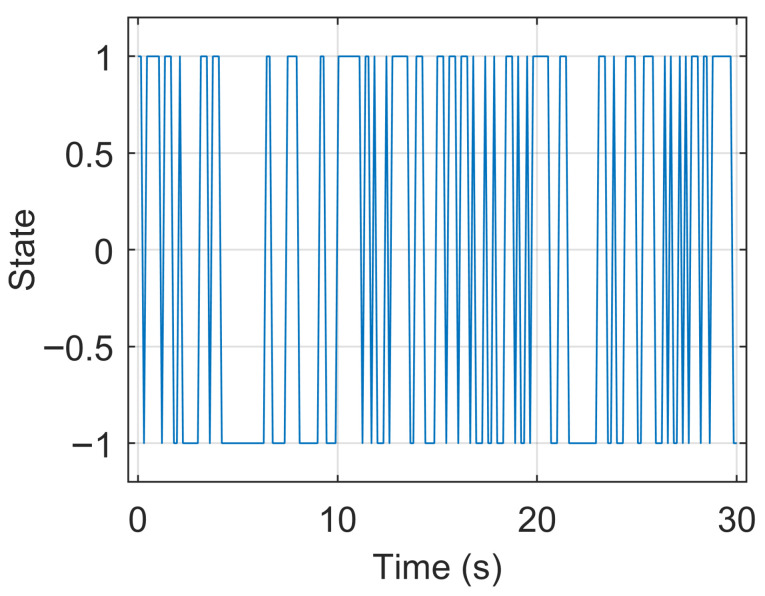
GBN used for the evaluation in the frequency domain, frequency range: 0.5–20 rad/s, duration: 30 s, $T_{sw}$: 0.15 s, corresponding $P_{ns}$: 0.58.

**Figure 7 sensors-25-05161-f007:**
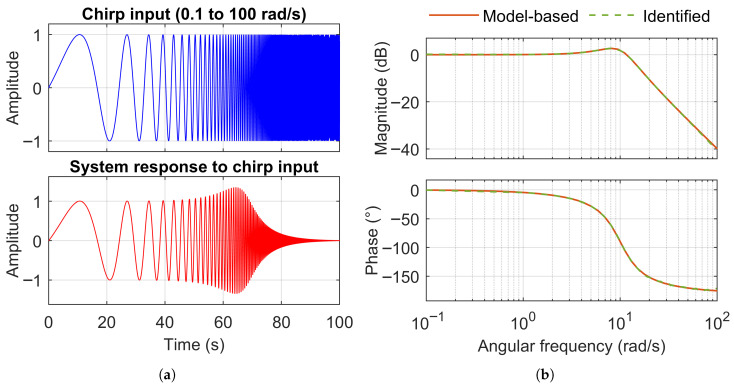
Response of a second-order system ($\omega_{n}=10\;\mathrm{rad}/s,\;\zeta =0.4$) to a chirp input applied across the full frequency range of interest (0.1 to 100 rad/s). (**a**) Time domain. (**b**) Frequency domain.

**Figure 8 sensors-25-05161-f008:**
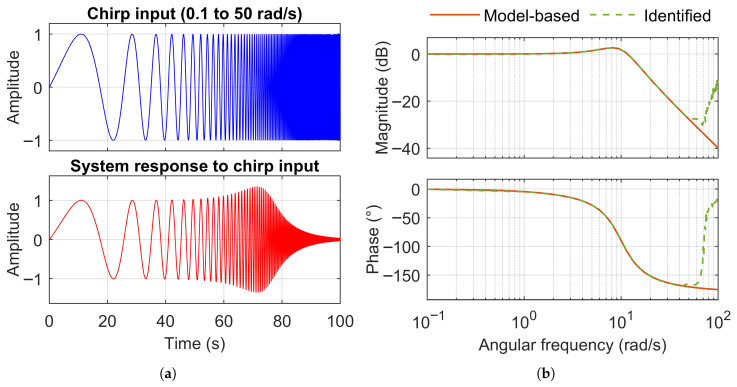
Response of a second-order system ($\omega_{n}=10\;\mathrm{rad}/s,\;\zeta =0.4$) to a chirp input applied across the shorter frequency range (0.1 to 50 rad/s) than the frequency range of interest (0.1 to 100 rad/s). (**a**) Time domain. (**b**) Frequency domain.

**Figure 9 sensors-25-05161-f009:**
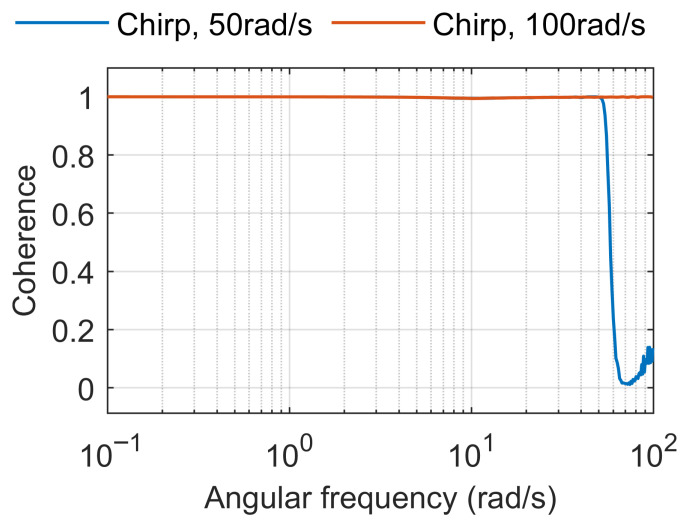
Comparison of the composite coherence obtained from a chirp input covering the full frequency range of interest (0.1 to 100 rad/s) versus a chirp input applied over a shorter frequency range (0.1 to 50 rad/s).

**Figure 10 sensors-25-05161-f010:**
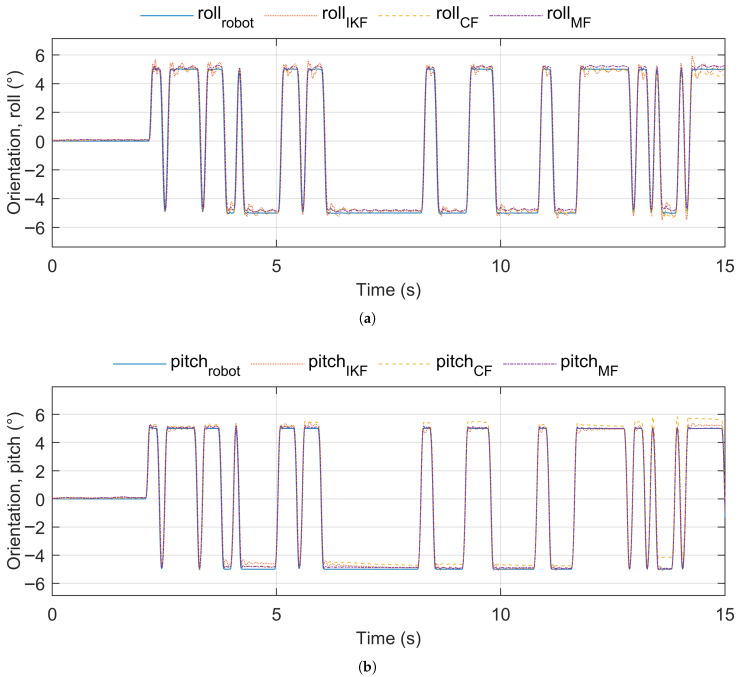
GBN test signal in the time domain (not the full duration of the signal is shown). (**a**) Applied to roll. (**b**) Applied to pitch.

**Figure 11 sensors-25-05161-f011:**
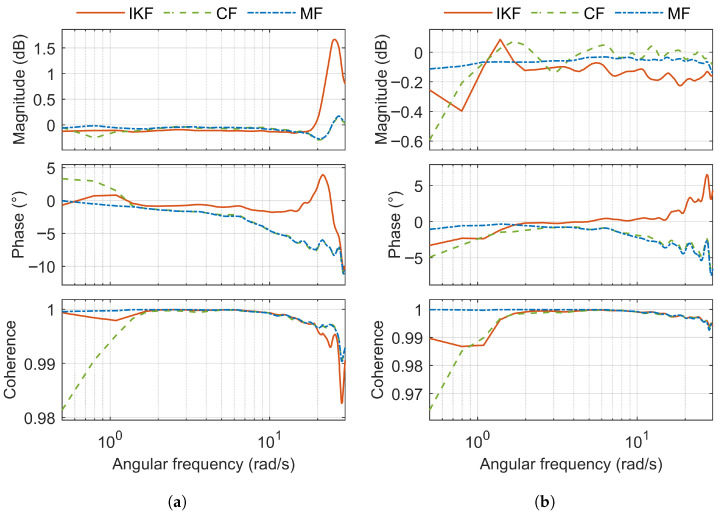
Composite frequency response and coherence for roll and pitch (GBN test signal is applied separately). (**a**) Roll. (**b**) Pitch.

**Figure 12 sensors-25-05161-f012:**
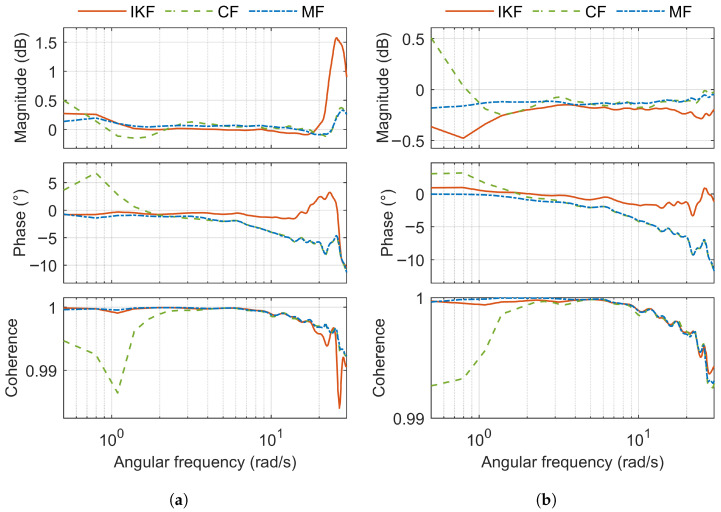
Composite frequency response and coherence for roll and pitch (GBN test signal is applied simultaneously for both). (**a**) Roll. (**b**) Pitch.

**Table 1 sensors-25-05161-t001:** Characteristics of Bosch MM7.10 IMU.

Parameter	Gyroscope	Accelerometer
Nominal measurement range	±163°/s	±41 m/s^2^
Cut-off frequency (−3 dB)	15 Hz	15 Hz
Sensitivity error	±2.0%	±2.0%
Non-linearity	±0.5°/s	±0.4 m/s^2^
Offset (bias)	±1.0°/s	±0.5 m/s^2^
Cross axis sensitivity	±1.5%	±1.5%
Output noise	0.2°/s RMS	0.04 m/s^2^ RMS
Resolution (absolute)	0.05°/s	0.022 m/s^2^

**Table 2 sensors-25-05161-t002:** Evaluation in the time domain.

Pose Sequence	Filter Type	RMSE|MaxAE Roll (°)	RMSE|MaxAE Pitch (°)	RMSE|MaxAE Sum (°)
Test Signal 1	IKF	2.07|5.62	0.99|2.41	3.06|8.03
	CF	0.68|1.75	0.56|1.34	1.24|3.09
	MF	0.46|1.54	0.40|1.05	0.86|2.59
Test Signal 2	IKF	2.14|6.22	1.08|5.58	3.22|11.80
	CF	1.88|4.60	1.17|3.23	3.05|7.83
	MF	1.79|4.84	1.18|3.86	2.97| 8.70
Test Signal 3	IKF	0.59|1.45	0.58|0.94	1.17|2.39
	CF	0.53| 1.11	0.54|0.86	1.07|1.97
	MF	0.55|1.23	0.57|0.97	1.12 |2.20

The lowest value for a given test signal and metric is marked in green, and the highest is marked in red.

**Table 3 sensors-25-05161-t003:** RMSE of the composite frequency response magnitude (RMSEM), RMSE of the composite coherence (RMSEC), and averaged equivalent time delay (AETD).

GBN Signal	Filter Type	RMSEM (dB)	RMSEC	AETD (ms)
Roll only	IKF	0.70	0.005	2
	CF	0.14	0.004	5
	MF	0.14	0.003	7
Pitch only	IKF	0.16	0.003	1
	CF	0.07	0.005	6
	MF	0.06	0.003	4
Roll and pitch	IKF	0.88	0.008	3
	CF	0.28	0.007	7
	MF	0.24	0.006	13

The lowest value for a given test signal and metric is marked in green, and the highest is marked in red. For the GBN test signal applied simultaneously to the roll and pitch axes, the reported metrics correspond to the sum of the individual metrics for roll and pitch.

## Data Availability

The data used for filters’ evaluation and tuning are available on request from the corresponding author.

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
