# Peer review of "Time and Frequency Domain Analysis of IMU-Based Orientation Estimation Algorithms with Comparison to Robotic Arm Orientation as Referenceâ€"

_sensors, 2025, doi:10.3390/s25165161_

Round 1
Reviewer 1 Report
Comments and Suggestions for Authors
Q1: In the abstract, you state that "for large amplitude signals ... MF shows the best performace". In terms of which metric is the performance compared?
Q2: The equations (2)-(3) use the symbol g_b for measured gravitational acceleration in body frame. However, the equation (9) use the symbol a_b. The relation between g_b and a_b needs to be clarified.
Q3: What is "decay factor for linear acceleration drift" mentioned on line 105? Add the prediction and output functions of ESEKF.
Q4: Please explain the procedure how the gain for the complementary filter was tuned. The value of 0.005 is suspicious, what is its accuracy?
Q5: What is the reason to use complementary filter on quaternions instead of computed Euler angles, which is more common?
Q6: Please explain the procedure how the learning rate beta for the Madgwick filter was tuned (how the value 0.006 was obtained).
Q7: Please explain what is the "rotation vector form" (line 173) and how it has been converted to Euler angles.
Q8: Please explain the choice of the experimental pose sequences (trajectories). By comparison of figures 1-3, it is hard to tell which one is "large-amplitude, slowly changing" and which is "small amplitude, fast changing" as mentioned in the abstract.
Q9: The abberviation MAE is commonly used for Mean Absolute Error, I would suggest to use different abberviation for maximum absolute error, e.g. MaxAE, to avoid confusion.
Q10: Please add the information about vendor-guaranteed accuracy and repeatability of the robotic arm at its TCP.
Q11: Please explain more deeply the intuition why you choose to evaluate the filter performance in the frequency domain.
Q12: Since the motion of any object is inherently smooth due to the object's inertia, it does not seem reasonable to evaluate the pose estimation filters on generalized binary signal. Please justify, why the sub-chapter 6.1.3 is needed in the paper.
Q13: Explain the choice of the parameters for the second order system in eq. (34). How it correlates to the evaluated pose estimation filters?
Q14: The composite coherence values in table 2 are very low. You mentioned that ideally the coherence should be equal to one. Hence it does not seem to be a good metric for evaluating the filters.
Q15: It is unclear how the experiments for frequency response were conducted. You mention the frequency up to 30 rad/s (almost 5 Hz), which is quite high value for a robot arm to follow. How can you guarantee the robot is actually following the defined reference trajectory at such high rate?
Q16: The figures are difficult to read. Please increase the font size for better accessiblity. Reconsider the color choice and scales in figure 10, there are too many series next to each other. There are too many pulses, please show smaller time interval.
Q17: The figure 9 shows the frequency response up to 100 rad/s. The horizontal axis need to be extended to show the response above the maximal chirp frequency.
Author Response
Thank you very much for taking the time to review our manuscript. Below, we provide detailed responses to your comments.
Responses:
Q1: In the abstract, you state that "for large amplitude signals ... MF shows the best performace". In terms of which metric is the performance compared?
R1: Metrics are RMSE and MaxAE for roll and pitch, and sum of RMSEs and sum of MAE for roll and pitch. Abstract is slightly rephrased for better clarity.
Q2: The equations (2)-(3) use the symbol g_b for measured gravitational acceleration in body frame. However, the equation (9) use the symbol a_b. The relation between g_b and a_b needs to be clarified.
R2: a_xb, a_yb, a_zb are the normalized components of the measured acceleration vector in the body-fixed frame. Normalization is done by the magnitude of the measured acceleration vector. In static conditions, this magnitude equals the gravitational acceleration g (meaning a_xb=g_xb/g, a_y=g_yb/g, a_z=g_zb/g). However, in dynamic conditions, the presence of external linear accelerations causes the measured acceleration magnitude to deviate from g, which can introduce errors in the orientation estimation.
Equations (2)-(3) basically describe purely static case.
Description in text is extended. Lines 69-72
Q3: What is "decay factor for linear acceleration drift" mentioned on line 105? Add the prediction and output functions of ESEKF.
R3: In MATLAB’s implementation of the IKF, the decay factor for linear acceleration drift is a scalar parameter in the range ([0, 1)), used to model the drift in linear acceleration as a low-pass-filtered white noise process. It controls how quickly the estimated linear acceleration bias decays over time. A lower value causes the filter to adapt more rapidly to changes in acceleration, which is suitable for highly dynamic motion. Conversely, a higher value assumes more slowly varying acceleration and results in a more persistent bias estimate.
Description of "decay factor for linear acceleration drift" (Lines 160-166) and Kalman equations are added (Equations 14-20).
Q4: Please explain the procedure how the gain for the complementary filter was tuned. The value of 0.005 is suspicious, what is its accuracy?
R4: All filters used in this work were tuned using a Generalized Binary Noise (GBN) signal with a wider frequency range than the test signal GBN. I added footnote in Section 3.1 (line 167) that redirects to GBN subsection
The default value of the filter gain is 0.01. The tuning was done by grid search (just going through different values of gain). Description is extended (line 189). I guess the resulting gain is lower than default one because GBN introduces high frequency components.
Q5: What is the reason to use complementary filter on quaternions instead of computed Euler angles, which is more common?
R5: Quaternion-based complementary filter from MATLAB’s Sensor Fusion and Tracking Toolbox was used which is described in paper: Valenti, R., I. Dryanovski, and J. Xiao. "Keeping a good attitude: A quaternion-based orientation filter for IMUs and MARGs." Sensors. Vol. 15, Number 8, 2015, pp. 19302-19330.
Overall, orientation representation using quaternions avoids singularities such as gimbal lock, which are associated with Euler angles, while providing rotation-order independence (due to a single rotation representation instead of three sequential ones) and ensuring better numerical stability and computational efficiency.
Then resulting quaternions were converted to Euler angles mainly because metrics were based on roll and pitch (yaw cannot reliably identified with 6-DoF IMU so it wasn’t included in metrics).
Q6: Please explain the procedure how the learning rate beta for the Madgwick filter was tuned (how the value 0.006 was obtained).
R6: Similarly to complementary filter gain, beta was tuned by simple grid search. Description of the tuning process is extended. (Line 223).
Q7: Please explain what is the "rotation vector form" (line 173) and how it has been converted to Euler angles.
R7: By "rotation vector form" Universal Robots refers to axis–angle representation. I added in the text that it means axis-angle representation (Lines 238-239).
https://www.universal-robots.com/articles/ur/application-installation/explanation-on-robot-orientation/#:~:text=Rotation%20Vector,can%20have%20a%20different%20orientation.
For conversion to other forms, MATLAB axang2quat and euler functions were used.
Q8: Please explain the choice of the experimental pose sequences (trajectories). By comparison of figures 1-3, it is hard to tell which one is "large-amplitude, slowly changing" and which is "small amplitude, fast changing" as mentioned in the abstract.
R8: All 3 pose sequencies are called large amplitude slowly changing. They are big amplitude compared to GBN signal used for frequency domain evaluation.
Note in the text is added to make it clearer (Line 260).
Q9: The abberviation MAE is commonly used for Mean Absolute Error, I would suggest to use different abberviation for maximum absolute error, e.g. MaxAE, to avoid confusion.
R9: Corrected.
Q10: Please add the information about vendor-guaranteed accuracy and repeatability of the robotic arm at its TCP.
R10: According to the manual, pose Repeatability per ISO 9283 ± 0.03 mm.
This information is added to the article (Line 231).
Q11: Please explain more deeply the intuition why you choose to evaluate the filter performance in the frequency domain.
R11: We chose to evaluate filter performance in the frequency domain to gain deeper insight into how each algorithm handles dynamic changes in orientation. While time-domain metrics like orientation error (RMSE, MaxAE) are effective for assessing accuracy during slow or stationary movements, they are not effective for dynamic motion. For example, a filter with a small time delay can show a large error even if it tracks motion well.
Frequency-domain analysis allows us to observe how filters attenuate or distort signals across a range of frequencies, quantify phase delays through the averaged equivalent time delay (AETD), and assess linear correlation for each frequency using coherence. This approach provides a more complete picture of filter behavior, especially under dynamic conditions.
Description in the text is extended (Lines 104-114).
Q12: Since the motion of any object is inherently smooth due to the object's inertia, it does not seem reasonable to evaluate the pose estimation filters on generalized binary signal. Please justify, why the sub-chapter 6.1.3 is needed in the paper.
R12: In this paper, GBN signal based on these papers is used:
Tulleken, H.J. Generalized binary noise test-signal concept for improved identification-experiment design. Automatica 1990, 48726, 37–49. https://doi.org/10.1016/0005-1098(90)90156-C.
Chen, J.K.; Yu, C.C. Optimal input design using generalized binary sequence. Automatica 1997, 33, 2081–2084. https://doi.org/10.1016/S0005-1098(97)00122-2.
Transition between states for GBN doesn’t happen immediately but with certain switching time. Switching time was empirically determined (e.g., for 5° amplitude GBN, it takes robot ~0.15 s to switch between -5° to 5° state).
However, that’s correct that velocity profile of theoretical GBN in mentioned papers is not smooth (moment when target state is reached velocity drops immediately to 0). And it’s smooth in real system which will to some degree distort the excitation range of real implementation of GBN. To mitigate that, robot was set to high acceleration to brake as fast as possible.
It is also worth noting that the frequency spectrum of GBN is not strictly confined to the interval omega_L to omega_H; rather, most of the signal power is concentrated in that region.
Q13: Explain the choice of the parameters for the second order system in eq. (34). How it correlates to the evaluated pose estimation filters?
R13: The second order system is not related to the orientation estimation filters. It was used as example to show that composite frequency response calculation method works properly. Because for this system we can get frequency response both analytically (because transfer function is known) and empirically (by comparing input/output signals with Ockier method)
Q14: The composite coherence values in table 2 are very low. You mentioned that ideally the coherence should be equal to one. Hence it does not seem to be a good metric for evaluating the filters.
R14: The values in the table are not composite coherences itself, but RMSE of coherence with one being reference coherence. For each frequency coherence error is calculated (coherence - 1) and then RMS is calculated.
Q15: It is unclear how the experiments for frequency response were conducted. You mention the frequency up to 30 rad/s (almost 5 Hz), which is quite high value for a robot arm to follow. How can you guarantee the robot is actually following the defined reference trajectory at such high rate?
R15: It was achieved by several measures: using small amplitudes for GBN (so transition between states (switching time) as short as possible), using high velocity (3m/s) and acceleration settings (15m/s2).
Switching time for GBN signals was determined from Real time data exchange (RTDE) that can provide robot data such as pose up to 500 Hz (100 Hz was used). From robot data (it was recorded to csv files), it was measured how much time it takes robot to go from high (e.g. 5°) to low state (e.g. -5°). Then using this information and frequency range of interest, GBN signal was calculated and executed.
For tuning GBN: frequency range 0.1-25 rad/s, duration of 60s, amplitude 3°, switching time 0.12s
For test GBN: frequency range 0.5-20 rad/s, duration of 30s, amplitude 5°, switching time 0.15s
No additional references other than RTDE were used.
Here is theoretical GBN used for testing (see Word attachment).
Here is the plot of robot orientation for roll GBN (see Word attachment).
It can be seen that robot manages to follow GBN quite closely.
Example of GBN for roll with 5° amplitude is also attached as video (see Word attachment).
Q16: The figures are difficult to read. Please increase the font size for better accessiblity. Reconsider the color choice and scales in figure 10, there are too many series next to each other. There are too many pulses, please show smaller time interval.
R16: Corrected.
Q17: The figure 9 shows the frequency response up to 100 rad/s. The horizontal axis need to be extended to show the response above the maximal chirp frequency.
R17: The idea was to show that the system needs to be excited within the region of interest in order to be correctly identified there. Of course, this means that if we go beyond the excitation region, the identification will not be accurate. This is already demonstrated by the chirp signal with a maximum of 50 rad/s (Figure 8). That’s why I believe it’s better to keep the frequency up to 100 rad/s, as that is the region of interest in the current example.

Reviewer 2 Report
Comments and Suggestions for Authors
Good day!
Interesting work.
Questions and comments:
1) The introduction lacks a critical review of the solutions in this area of ​​research. The authors immediately began the text with a problem. It is difficult for the reader.
2) For ease of reading and orientation in the text, add the structure of the article at the end of the introduction
3) Figures 2-4 are impossible to read. Similarly, improve the quality of the oscillograms in the text.
Best regards, reviewer
Author Response
Thank you very much for taking the time to review our manuscript. Below, we provide detailed responses to your comments.
Responses:
1) The introduction lacks a critical review of the solutions in this area of ​​research. The authors immediately began the text with a problem. It is difficult for the reader.
The introduction is extended (Lines 83-136).
2) For ease of reading and orientation in the text, add the structure of the article at the end of the introduction
The structure of the article is added at the end of the introduction (Lines 137-141).
3) Figures 2-4 are impossible to read. Similarly, improve the quality of the oscillograms in the text.
The quality of figures is improved.
Reviewer 3 Report
Comments and Suggestions for Authors
Review – Sensors-3716396 (Sultan & Greiser).
A well written paper, easy to read and understand. As far as I can tell, the mathematical analysis is correct. The English is on the whole excellent, though a few small errors I have noted below.
Line 222: “Important part of the evaluation in the frequency domain is Fourier Transform which
allows to get frequency spectrum of the signal.” Is German. The correct translation is: “An important part of the evaluation in the frequency domain is the Fourier Transform which provides the frequency spectrum of the signal.”
Similarly, line 224: “ The discrete-time fourier transform (DTFT) allows to analyzes the frequency content of infinite-length discrete-time signals.” should read “ By means of the discrete-time Fourier transform (DTFT) , the frequency content of infinite-length discrete-time signals can be analysed.”
Line 272: should read: “...for tuning filter parameters.”
Line 300: should read: “For filter evaluation in the frequency domain,”
Line 343: “Chirp-Z tranform” should read “Chirp-Z transform”
The widespread use of the apostrophe appears to be somewhat erroneous. Please consult <https://dictionary.cambridge.org/de/grammatik/britisch-grammatik/apostrophe> for correct usage.
One small tip: I know its frequently used today but “setup” is technical slang (lines 82, 86, 163, 168, figure 1). Better to use “configuration”, “apparatus” or “equipment”.
Conclusion: accept with very minor corrections.
Author Response
Thank you very much for taking the time to review our manuscript. Below, we provide detailed responses to your comments.
Line 222: “Important part of the evaluation in the frequency domain is Fourier Transform which allows to get frequency spectrum of the signal.” Is German. The correct translation is: “An important part of the evaluation in the frequency domain is the Fourier Transform which provides the frequency spectrum of the signal.”
Corrected (Lines 298-299).
Similarly, line 224: “ The discrete-time fourier transform (DTFT) allows to analyzes the frequency content of infinite-length discrete-time signals.” should read “ By means of the discrete-time Fourier transform (DTFT) , the frequency content of infinite-length discrete-time signals can be analysed.”
Corrected (Lines 300-301).
Line 272: should read: “...for tuning filter parameters.”
According to the provided link (https://dictionary.cambridge.org/de/grammatik/britisch-grammatik/apostrophe), when a noun is plural, the apostrophe is placed after the s. In this case, "filters" is plural, so the correct form should be "filters'", shouldn't it?
Line 300: should read: “For filter evaluation in the frequency domain,”
According to the provided link (https://dictionary.cambridge.org/de/grammatik/britisch-grammatik/apostrophe), when a noun is plural, the apostrophe is placed after the s. In this case, "filters" is plural, so the correct form should be "filters'", shouldn't it?
Line 343: “Chirp-Z tranform” should read “Chirp-Z transform”
Corrected (Line 423).
The widespread use of the apostrophe appears to be somewhat erroneous. Please consult <https://dictionary.cambridge.org/de/grammatik/britisch-grammatik/apostrophe> for correct usage.
One small tip: I know its frequently used today but “setup” is technical slang (lines 82, 86, 163, 168, figure 1). Better to use “configuration”, “apparatus” or “equipment”.
Thank you for your helpful comment regarding the use of the term “setup.” While we understand that “setup” can be considered informal in some contexts, it is also widely accepted in engineering and experimental literature to describe the physical arrangement of hardware and components. In this manuscript, we use the term “experimental setup” specifically to refer to the physical combination of the robotic arm, IMU, and data acquisition interfaces.
For reference, similar usage can be found in the articles with similar topic published in MDPI Sensors, such as:
- https://www.mdpi.com/1424-8220/21/20/6858
- https://www.mdpi.com/1424-8220/24/24/8179
Therefore, we would prefer to retain the term “setup” in its current context. However, if the editorial team strongly recommends a more formal alternative (e.g., “apparatus” or “configuration”), we are happy to revise the text accordingly.
Reviewer 4 Report
Comments and Suggestions for Authors
1. Figure 4: What causes the dips in the graphs (29 and 40 seconds)?
2. Lines 286-287: the authors write "In this work, it is achieved by using small amplitudes and moving the robot with maximum velocity and acceleration". What are the characteristics of the robot? With what maximum speed and acceleration is it possible to move?
3. In Figure 10, perhaps it is worth enlarging the areas of interest to estimate the duration of the transient process in each filter?
4. It is necessary to clarify whether the Bosch MM7.10 inertial module was pre-calibrated? It is worth providing its accuracy characteristics for measuring accelerations and angular velocities.
5. Since the readings of the inertial module depend on temperature, it is necessary to clarify whether the experimental studies were carried out at room temperature. If so, how well will the filters work when it changes?
Author Response
Thank you very much for taking the time to review our manuscript. Below, we provide detailed responses to your comments.
1. Figure 4: What causes the dips in the graphs (29 and 40 seconds)?
It appears that the robot was unable to change its orientation from RPY (roll–pitch–yaw) = 0–0–0° to RPY = 45–45–0° without introducing a slight change in yaw. Therefore, these dips are not caused by the filters, but rather by the robot itself.
2. Lines 286-287: the authors write "In this work, it is achieved by using small amplitudes and moving the robot with maximum velocity and acceleration". What are the characteristics of the robot? With what maximum speed and acceleration is it possible to move?
For GBN program, the following settings were chosen: tool speed 3m/s; tool acceleration 15 m/s^2. Footnote (Line 363) is added.
According to the datasheet: max TCP speed 4 m/s. This information is added to the article (Lne 232).
3. In Figure 10, perhaps it is worth enlarging the areas of interest to estimate the duration of the transient process in each filter?
Corrected.
4. It is necessary to clarify whether the Bosch MM7.10 inertial module was pre-calibrated? It is worth providing its accuracy characteristics for measuring accelerations and angular velocities.
IMU was calibrated at the factory. Table with IMU characteristics is added to the article (Table 1).
5. Since the readings of the inertial module depend on temperature, it is necessary to clarify whether the experimental studies were carried out at room temperature. If so, how well will the filters work when it changes?
The experiments were conducted only at the room temperature. This information is added to the article (Lines 244-245).
Round 2
Reviewer 1 Report
Comments and Suggestions for Authors
The revised version of the article is a signifficant improvement. Authors addressed most of my previous concerns. Here are my comment regarding the readability and novelty of the paper:
Q1: The figures 10-11 show the time-domain signals (ground truth vs. estimated state). It is clear that the estimation errors are caused mainly by the constant delay of the filters. I would suggest authors to show the errors of the filters in time domain by comparing the estimated values with time-delayed ground truth (using AETD value as a delay). It would better support their claim that "filter with a small time delay can show a large error even if it tracks motion well".
Q2: Is there any intuition why the AETD values are signifficantly different for "roll only" and "pitch only" scenarios? Please explain in the paper.
Author Response
Thank you for your feedback.
During further analysis, we discovered that the dominant contributor to the AETD values was the time offset between IMU and robot data , despite both being timestamped using the recording computer's system clock. This mismatch is primarily caused by transmission and buffering delays, for example, USB buffering in the PCAN-USB adapter, which introduce latency in the IMU data stream relative to the robot data.
To address this, we estimated and compensated for the time shift between the IMU and robot data by minimizing the Averaged Equivalent Time Delay (AETD). This was done based on the composite frequency response phase, using the time-shifted robot orientation as input and the integrated gyroscope measurements as output. The time shift that minimized the AETD was selected and applied. The estimated and compensated shifts for the pose sequence tests were as follows: signal 1 — 31 ms; signal 2 — 16 ms; signal 3 — 20 ms. For the Generalized Binary Noise (GBN) test signals: GBN roll — 26 ms; GBN pitch — 82 ms; GBN roll and pitch — 14 ms.
This compensation explains the initially large differences in AETD values between the "roll only" and "pitch only" cases. After time alignment, the AETD values became significantly more consistent across all test scenarios. These changes are now reflected in the updated plots and results in the manuscript, and the delay compensation procedure has been described in the revised Experimental Setup section (Lines 252-259).